# Rapid Limit Test of Eight Quinolone Residues in Food Based on TLC-SERS, a New Limit Test Method

**DOI:** 10.3390/molecules28186473

**Published:** 2023-09-06

**Authors:** Honglian Zhang, Min Zhang, Li Li, Wei Dong, Qiyong Ren, Feng Xu, Yuanrui Wang, Tao Xu, Jicheng Liu

**Affiliations:** 1School of Pharmacy, Qiqihar Medical University, Qiqihar 161006, China; zhanghonglian_2006@163.com (H.Z.); 17382849773@163.com (M.Z.); pingguoweiweiwei@126.com (W.D.); 13856106964@163.com (Q.R.); 15845205504@qmu.edu.cn (F.X.); harvey-333@163.com (T.X.); qyybliu@126.com (J.L.); 2Qiqihar Institute for Food and Drug Control, Qiqihar 161006, China; wangyuanrui2021@163.com

**Keywords:** TLC-SERS, quinolones, residues, aquatic products, animal foods

## Abstract

Residual quinolones in food that exceed their maximum residue limit (MRL) are harmful to human health. However, the existing methods used for testing these residues have limitations; so, we developed a new limit test method called TLC-SERS to rapidly determine the levels of residues of the following: enrofloxacin (A), ciprofloxacin (B), ofloxacin (C), fleroxacin (D), sparfloxacin (E), enoxacin (F), gatifloxacin (G), and nadifloxacin (H). The residues ware preliminarily separated via TLC. The tested compounds’ position on a thin-layer plate were labeled using their relative R_f_ under 254 nm ultraviolet light, and an appropriate amount of nanometer silver solution was added to the position. The silver on the plate was irradiated with a 532 nm laser to obtain the SERSs of the compounds. The results show significant differences in the SERS of the eight quinolones: the LODs of H, A, D, E, C, G, F, and B were 9.0, 12.6, 8.9, 19.0, 8.0, 8.7, 19.0, and 12.6 ng/mL, respectively; and the RSD was ≤4.9% for the SERS of each quinolone. The limit test results of 20 samples are consistent with those obtained via UPLC–MS/MS. The results indicate that TLC-SERS is a specific, sensitive, stable, and accurate method, providing a new reference for the rapid limit test of harmful residues in foods.

## 1. Introduction

Since the early 2000s, quinolones, as veterinary drugs with antibacterial effects, have been widely used in animal husbandry and liquid product industries. These quinolones mainly include enrofloxacin (A), ciprofloxacin (B), ofloxacin (C), fleroxacin (D), sparfloxacin (E), enoxacin (F), gatifloxacin (G), and nadifloxacin(H) [1,2,3,4,5]. Levels of quinolones in aquatic products and other animal foods exceeding the maximum residue limit (MRL) cause serious harm to consumers’ health [6]. Notably, the levels of quinolones commonly exceed their MRL in food. For example, according to the relevant literature, in 2020, enrofloxacin, ciprofloxacin, and ofloxacin were found to exceed their MRL in some aquatic products, eggs, and pork [6,7,8,9]. To prevent this from occurring, a rapid limit test method must be established for controling the residues in these foods.

In China’s national food standards, the MRLs of these eight quinolones (A, B, C, D, E, F, G, and H) in food are specified as 100.0, 100.0, 2.0, 5.0, 5.0, 5.0, 5.0, and 5.0 μg/kg, respectively; the levels of these residues are usually determined via UPLC-MS/MS. Despite its high sensitivity and specificity in quantitative and qualitative analyses, UPLC-MS/MS still requires complicated sample pretreatment procedures as well as hours or even days for the completion of the whole analysis process [2,3,4,5,6].

Traditional thin-layer chromatography (TLC) is a fast and convenient method for separating chemical components from complex matrices. With the help of auxiliary means, such as chemical color development and ultraviolet light irradiation, this method only reflects the chemical structural characteristics of a certain functional group of the testing component, and its sensitivity is relatively low [10,11,12,13], so it cannot be directly used for limit tests of residues in food.

Analysis based on Raman spectroscopy is a new method that can be used for the rapid detection of the chemical compositions of food. The results of Raman spectroscopy show the inelastic scattering of compound molecules after being irradiated using a laser, so the fingerprint structure information of the constituent compounds can be reflected by Raman spectroscopy. Additionally, Raman spectroscopy is almost unaffected by water and silica gel under experimental conditions. Although this method is highly specific, its sensitivity is relatively low. To meet the requirements for the analysis of some trace components, the sensitivity can be increased by more than 10^4^ times through surface-enhanced Raman spectroscopy (SERS) [10,11,12,13]. SERS technology provides molecular fingerprints with high-specificity and high-sensitivity spectral features and has been widely applied in the field of analysis. For example, SERS technology can be used for the specific detection of quinolone residues when harmful residues have been preliminary separated in dietary supplements or food using TLC [14,15,16,17,18,19,20].

The purpose of this study is to develop a new method for testing if compounds in food are under the limits by combining TLC with SERS, called TLC-SERS. TLC-SERS is a semiquantitative analysis method that falls between qualitative and quantitative analyses [21]. In TLC-SERS, when the SERS from the sample solution is the same as the SERS from the reference solution, their characteristic peak heights are compared to determine if the residue in the sample exceeds its MRL. If the peak height of the sample solution is larger than that of the reference solution, the level of the residue in the food exceeds its MRL, and the quality of the sample is unacceptable. If the peak height of the sample solution is less than or equal to that of the reference solution, the level of the residue in the food does not exceed its MRL, and the quality of the sample is acceptable.

Comparing TLC-SERS with UPLC-MS/MS, their sensitivity and specificity are similar; however, the methods differ widely in the time and cost required to complete the same experiment (Appendix A). In TLC-SERS, because the matrix in the food is composed of some compounds that do not produce the SERS, the sample pretreatment is simple, the method is easy to operate, and less time is required (approximately 15 min); so, the method is suitable for the rapid on-site detection of contaminants in food. However, with UPLC-MS/MS, to prevent the chromatographic column being blocked by the food matrix, a cumbersome and time-consuming (approximately 250 min) sample pretreatment method is required, so the method is unsuitable for the rapid on-site detection of food contaminants. In addition, TLC-SERS can be performed with small, low-cost, and portable Raman spectroscopy instruments, whereas UPLC-MS/MS must be completed using large and expensive instruments. Therefore, TLC-SERS is more suitable for on-site analysis than UPLC-MS/MS.

The residues of the eight quinolones in foods can be rapidly separated and specifically detected by TLC-SERS; so, the method provides a new reference basis for the rapid on-site analysis of harmful residues in food.

## 2. Results and Discussion

### 2.1. Characterization and Stability of the SERS of Active Substrates

The special nature of the SERS of the active substrates was characterized as follows: By diluting the nanometer silver solution 14 times, the ultraviolet absorption spectrum was obtained via detection using the diluted solution, which showed the maximum absorption peak at 424 nm (Figure 1a). The appearance of the substrates was characterized as a ball shape (Figure 1b).

The particle size and zeta potential of the substrates were measured at different time points (0~21 days), with most of the particles being 41.40~43.14 nm (Figure 1c,d) and the zeta potential being 31.14~35.80 mv (Figure 1e). Figure 1c–e confirms that the SERS of the active substrates were uniformly distributed and stable.

### 2.2. Relative R_f_ and Raman Spectra

Using TLC, these quinolones were effectively separated, except for C and E, in the mixed reference substance solution 1 (Figure 2a). Using fleroxacin as a reference, the relative R_f_ of the main spot was 1.22 (H), 1.16 (A), 1.00 (D), 0.90 (E), 0.89 (C), 0.74 (G), 0.56 (F), and 0.44 (B). When an appropriate amount of the above reference solution was placed into an liquid product without any quinolones, the tested solution was prepared by following the sample solution preparation method. The above TLC was repeated, with the solution displaying the same relative R_f_ on the thin-layer chromatogram (Figure 2b). When the liquid product was used instead of animal food, the same result was obtained. This indicated that the matrix in the food had no effect on the relative R_f_ of the quinolones.

On the chromatogram (Figure 2b), the eight quinolones separated were separately enriched to higher concentrations via in situ chromatography. Then, the Raman spectra of the enriched quinolones was directly measured (Figure 2c).

### 2.3. SERS of Quinolones on TLC

With TLC-SERS, we found an obvious spot on the thin-layer chromatogram of the fleroxacin reference substance solutions, and the R_f_ was measured as 0.70 (Figure 3a). The mixed reference substance solution 2 produced no spot on the chromatogram; so, the position of the eight quinolones were indicated by their R_f_. This R_f_ was calculated using their relative R_f_ and the R_f_ of fleroxacin; so, the R_f_ values were 0.85 (H), 0.81 (A), 0.70 (D), 0.63 (E), 0.62 (C), 0.52 (G), 0.39 (F), and 0.31 (B). A nanometer silver solution was added dropwise to the positions of the eight R_f_ mentioned above. Figure 3b shows that the SERS of the eight quinolones (H, A, D, E, C, F, G, F, and B) were obtained from the chromatogram of the mixed solution 2, but almost no Raman signal was observed from the corresponding blank positions. This indicates that the nanometer silver solution had no effect on the SERS of the quinolones.

### 2.4. Determination of EFs

The mass (M_blank_) and the corresponding characteristic Raman spectral peak intensity (I_blank_) of the quinolones are shown in Figure 2b,c, respectively. The mass (M_SERS_) and the corresponding characteristic SERS peak intensity (I_SERS_) of the same quinolones are shown in Figure 3a,b, respectively. The Raman enhancement factor (EF) was calculated as EF = (I_SERS/_M_SERS_)/(I_blank_/M_blank_), where I_SERS_ and I_blank_ are the Raman intensities at the characteristic peaks of the SERS of the active substrate and blank substrate, respectively; and M_SERS_ and M_blank_ are the mass (µg) of the quinolones on the SERS of the active substrate and the blank substrate, respectively. The measurements show that the EFs of eight quinolones were in the range of 1.1 × 10^4^~3.1 × 10^6^ (Table 1 and Appendix A).

To increase the sensitivity of the detection of the quinolones, we also prepared nanometer gold solutions as the active SERS substrates. The experimental results show that their enhancement effect on the Raman characteristic peaks of the eight quinolones was not as strong as that of the nanometer silver solutions, producing an EF of 1.1 × 10^4^~1.1 × 10^5^.

### 2.5. Comparison of Raman Spectroscopy and SERS

Using the aforementioned apparatus and conditions, the Raman spectra of the eight quinolone reference substances were separatelydetected, with eight spectra showing significant differences except for the blank with no signal (Figure 2c). The SERS diagram of the substances are shown in Figure 3b. When the SERS of the quinolones was compared with their Raman spectra, the results show that the shape and relative intensity of the main characteristic peaks were notably changed, but the Raman shifts (cm^−1^) of the peaks were similar. In addition, the number of characteristic peaks was lower in the SERS than in the Raman spectra (Table 1). This result indicates that the SERS of the quinolones strongly correlated with their Raman spectra, demonstrating that the SERS reflected some structural information of the quinolones. Therefore, TLC-SERS can be used as a specific method for analyzing quinolones.

### 2.6. Identification of Quinolones by Combining SERS and Relative R_f_

As shown in Table 1, for the different quinolones, the relative R_f_ values differed in the TLC, as did the relative strength and number of characteristic peaks in the SERS. The results of the experiment show that the quinolones could be identified by combining their relative R_f_ with the SERS results. This method has a higher specificity and selectivity. We provide some examples below.

Due to the close relative R_f_ of nadifloxacin (H) and enrofloxacin (A) (1.22 and 1.16, respectively; Table 1), the separation of the two components was relatively small, hindering their identification using TLC solely. Carefully observing H and A in the SERS results (Figure 3b) and their characteristic peaks (Table 1), we found eight peaks (H) and four peaks (A) with Raman shifts (1610~1157 cm^−1^). Additionally, the relative peak intensities were 0.57 (H) and 0.77 (A) from β_CH2_ or β_CH3_; the relative peak intensities were 0.32 (H) and 0.65 (A) from ν_C-N_. This result indicates that, although the differences in the two components’ R_f_ was small according to TLC, the TLC-SERS spectra were significantly different. Therefore, we established a new analytical method by combining the relative R_f_ of the two compounds (H and A) with their characteristic SERS values, and this method more accurately distinguishes H from A.

Due to the almost identical relative R_f_ of sparfloxacin (E) and ofloxacin (C) (0.90 and 0.89, respectively; Table 1), the two components could not be effectively separated solely through TLC. Carefully observing the SERS of E and C (Figure 3b) and their characteristic peaks (Table 1), we found remarkable differences. By comparing the chemical structures of the two components, we found an additional CH_2_ in E, which led to stronger in-plane binding vibrations (β_CH2_ and β_CH3_) in the SERS. The relative peak intensities of the peak (β_CH2_ and β_CH3_) were 1.03 (E) and 0.68 (C) (Table 1). In addition, we found an additional NH_2_ in E, which led to a stronger stretching vibration (ν_C-N_) in the SERS. The relative peak intensities were 0.71 (E) and 0. 57 (C). This result indicates that, although C and E cannot be effectively separated by TLC, the SERS of E and C could be separately obtained with TLC-SERS, and the SERS values of the two were completely different. Therefore, we established a new analytical method by combining the relative R_f_ of the two compounds (E and C) with their characteristic SERS to accurately distinguish E from C.

### 2.7. Experiment with a Simulated Positive Sample

When liquid samples without quinolones were used as the negative samples, an appropriate amount of the eight quinolones was placed into the negative samples to prepare the simulated positive sample, making the content of quinolones to be equal to the MRL of quinolones in food. This means that the content of quinolones (A, B, C, D, E, F, G and H) in the simulated positive sample was 100.0, 100.0, 2.0, 5.0, 5.0, 5.0, 5.0, and 5.0 μg/kg, respectively.

According to the preparation method of the sample solution, an appropriate amount of the simulated positive samples were added to anhydrous ethanol to prepare a simulated positive sample solution. We prepared the negative sample solution using the same method. Then, 10.0 µL of the reference substance solutions, the simulated positive sample solutions, and the negative sample solution were deposited onto the GF_254_ thin-layer plate. The experiment was conducted by TLC-SERS, and the result is shown in Appendix A, which indicates that the SERS of the simulated positive samples is in accordance with the corresponding reference substance when no Raman signal is obtained in the negative sample. The result further confirms that the matrix in the aquatic products did not interfere with the limit test of the eight quinolone residues, showing that TLC-SERS has a strong specificity and selectivity.

### 2.8. Stability and Feasibility Test

Using TLC-SERS, over 20 days, 10 groups of SERS diagrams were obtained via measuring the mixed reference substance solution 2 with the same nanosilver solution (the surface-active substrate) every other day. Taking quinolone (H) as an example, the relative standard deviations (RSDs) of the heights of the four characteristic peaks (ν_C=C,_ *β*_CH2_*, β*_CH3*,*_ and ν_C-N_) were detected as 4.6%, 5.1%, 3.8%, and 6.5%; the range was 3.8%~6.5% (H). Similarly, the RSDs of the characteristic peaks of the other quinolones were 3.2~6.8% (A), 3.9~6.1% (D), 3.9~6.7% (E), 4.1~7.9% (C), 4.3~6.5% (g), 4.1~6.6% (F), and 5.1~6.9%. The results show that the stability of the method was reliable when using a surface-active substrate that was stored for 20 days.

After placing the above-simulated positive sample solutions in TLC-SERS for 0, 1, 2, 4, or 8 h, the SERS values of the eight quinolones in the solutions at different times were sequentially obtained via TLC-SERS, and the RSDs of the heights of the four characteristic peaks (*ν*_C=C_, *β*_CH2_, *β*_CH3_, and *ν*_C-N_) were detected as 1.9~2.2% (H), 2.6~3.3% (A), 2.2~3.5% (D), 3.5~3.7% (E), 4.1~4.9% (C), 3.6~3.9% (G), 3.2~3.5% (F), and 3.1~3.4% (B), indicating that the quinolone solution was relatively stable over eight hours.

We replaced the DXR™xi Raman Imaging Microscope used in this experiment with a Thermo Truscan RM Hand-held Raman Spectrometer and repeated the above operation. The results show that the results of the quinolone determination with the two instruments were basically the same, indicating that this method can be used for rapid detection in the field using a hand-held instrument.

### 2.9. Limit of Detection Test

With the proposed method, a gradient concentration range of the solution should be used to determine the limit of detection (LOD), including the MRL (ng/mL) of quinolones in food. For example, the MRL of nadifloxacin (H) in food is 5.00 μg/kg, which is equivalent to 20 ng/mL of nadifloxacin in the sample solution. The concentration range of the reference substance (H) was 8.5~50.0 ng/mL. Similarly, Table 2 shows that the concentration ranges of the reference substances (A, D, E, C, G, F, and B) were 12.5~420.0, 8.5~50.0, 12.5~50.0, 8.0~32.0, 8.5~50.0, 12.5~50.0, and 12.5~420.0 ng/mL, respectively.

Using TLC-SERS, 10.0 µL of different concentrations of the solution were separately deposited onto GF_254_ thin-layer plates, and their corresponding SERS values were obtained. The results are shown in Appendix A. A signal-to-noise ratio of 3:1 (S/N = 3:1) was considered as the LOD of quinolones. The S/N was calculated by the ratio of the height of the quinolones’ characteristic peak (*ν*_C-N_) to that of the blank noise peak, and the Raman shifts of these characteristic peaks (*ν*_C-N_) occurred at 1157 (H), 1180 (A), 1237 (D), 1175 (E), 1151 (C), 1151 (G), 1170 (F), and 1151 (B) cm^−1^. The S/N values of the characteristic peak (ν_C-N_) of the quinolone reference materials at different concentrations were separately determined using TLC-SERS, and the measurement was repeated three times for each peak. The RSDs of the S/N of the eight quinolones were 2.9~4.3% (H), 2.2~3.8% (A), 2.9~4.1% (D), 2.9~3.7% (E), 3.1~4.9% (C), 3.0~4.5% (G), 3.1~4.6% (F), and 2.1~3.9% (B). This result indicates that this method has a good repeatability.

The curves were established by the quinolone concentration and S/N. The results are shown in Figure 4. According to the curves, the LODs of the eight quinolones are shown in Table 2.

As preparing the sample solution, the quinolone residues were extracted from 2 g (0.002 kg) of food and were diluted with 500.0 μL (0.5 mL) of anhydrous ethanol. The method used for converting MRL from μg/kg to ng/mL in Table 2 is as follows:MRL (ng/mL) = MRL (μg/kg) × 0.002 kg × 1000/0.5 mL = 4 × MRL (µg/kg)

In Table 2, which compares the LOD with the corresponding MRL (ng/mL), the results show that the LOD of each quinolone was not larger than its MRL. Therefore, the sensitivity of the method (TLC-SERS) meets the requirements for testing whether the levels of the eight quinolone residues meet the limits in food.

Notably, the heights of the characteristic peaks in the SERS increased with the quinolone concentration when the quinolone concentration was above its LOD (Figure 4). If quinolone residues are present in food, we can determine whether the amount of the residue exceeds its MRL via a comparison of the heights of the characteristic peaks of quinolones in food with those of the corresponding references in the mixed reference substance solution 2. In the mixed reference substance solution 2, the concentrations of quinolones were equal to their corresponding MRL (ng/mL), as shown in Table 2.

### 2.10. Limit Test Using Real Samples

Ten batches of different varieties of aquatic products and ten batches of different animal foods were taken separately (Table 3) The corresponding twenty sample solutions were prepared in accordance with the method described in Section 3.5. Using TLC-SERS (Section 3.7), 10.0 μL each of the fleroxacin (D) reference substance solution and mixed reference substance solution 2 as well as twenty batches of the sample solutions were separately dropped onto the same GF_254_ thin-layer plate. The detection results show that no SERS signal was observed in any sample, except for samples 3 and 8. The SERS values of samples 3 and 8 were the same as those of enrofloxacin (A) and ciprofloxacin (B) in the mixed reference substance solution 2, respectively. Figure 5 shows that the characteristic peaks of sample 3 were higher than those of the reference substance A, and the characteristic peaks of sample 8 were weaker than those of the reference substance B. This finding indicates that the content of enrofloxacin (A) exceeded the MRL (100 µg/kg) in sample 3, and the quinolone contents (A, B, C, D, E, F, G, and H) did not exceed their corresponding MRL in the other samples. That is, according to China’s national food standards, of the twenty batches of samples tested using TLC-SERS, except for enrofloxacin (A) in sample 3, the levels of the eight quinolones in the other samples were all below the limit.

To verify the accuracy of TLC-SERS, UPLC–MS/MS was used to quantitatively determine the level of the eight quinolones in the twenty batches of samples. The results show that the data obtained using the two methods were fully consistent, which indicates that TLC-SERS is accurate and reliable. The chromatography of the eight quinolones is shown in Appendix A_._ The chromatography results of samples 3 and 8 are shown in Appendix A, respectively.

## 3. Materials and Methods

### 3.1. Materials

All reagents were of analytical grade and purchased from Merck Drugs and Co., Darmstadt, Germany. The reference substances of enrofloxacin (98%), ciprofloxacin (82.1%), ofloxacin (99.7%), fleroxacin (99.2%), sparfloxacin (99.4%), enoxacin (91.5%), gatifloxacin (97.2%), and nadifloxacin (97%) were purchased from the China Food and Drug Control Institute (labeled as A, B, C, D, E, F, G, and H, respectively), and anhydrous ethanol (99.5%) was used to dissolve these quinolone compounds. We used ten batches of real samples of aquatic products, which were obtained from five different manufacturers (China), and the ten real samples of the other animal foods were supplied by other manufacturers (China). Anhydrous sodium sulfate was used to remove protein from the food, and acetonitrile (99.5%) was used to extract the quinolone residues from the food. Dichloromethane (99.5%) and methanol (99.5%) were used as developing agents in TLC. Silver nitrate and sodium citrate were used to prepare the active SERS substrates.

TLC was performed using a thin-layer plate (Merck KGaA, Darmstadt, Germany) composed of high-performance silica gel and fluorescing additive F_254_. The plate is called a GF_254_ thin-layer plate and had a layer thickness of 0.2 ± 0.03 mm, a particle size of 8 ± 2 μm, and an aluminum carrier. The microinjector (10 μL) used for spotting on the thin-layer plates was purchased from Zhenhai Glass Instrument Factory, Ningbo, China.

### 3.2. Apparatus and Conditions

The SERS or Raman or Raman spectra of the quinolone components were obtained with a DXR™ xi Raman Imaging Microscope (Thermo Fisher Scientific, Waltham, MA, USA), with a laser excitation wavelength of 532 nm, a resolution of 5.0 cm^−1^, and a 10× long working distance objective. The excitation power was 10 mW, the integration time was 0.5 s, and the number of scans was 20. The scan range was 3300–100 cm^−1^, with a 50 μm confocal pinhole DXR532 full-range grating (400 line/mm). The detector was a TE-cooled electron-multiplying CCD (EMCCD). Point scanning was chosen as the scanning mode.

An ultraviolet analyzer (YOKO-2F, Wuhan YOKO Technology Ltd., Wuhan, China) was used to mark the principal spot on the TLC under a 254 nm wavelength. An ultraviolet visible spectrophotometer (T_6_, Beijing Puxi General Instrument Co., Ltd., Beijing, China) was used to detect the ultraviolet absorption spectrum of the active SERS substrates. A transmission electron microscope (HT 7700, Beijing Shengjiachen Ke & Trade Co., Ltd., Beijing, China) was used to characterize the particle appearance of the substrates. A nanoparticle size analyzer (Nicomp 380 ZLS, Shanghai of meijia Co., Ltd., Shanghai, China) was used to measure the particle size and the particle zeta potential.

Ultra-high-performance liquid chromatography tandem mass spectrometry (UPLC–MS/MS) was operated on a Dionex UltiMate 3000 ultra-performance liquid chromatography tandem triple fourth stage rod composite linear ion trap mass spectrometer system (AB Sciex QTRAP 6500, AB SCIEX.com, accessed on 10 September 2020. Santa Clara, CA, USA). The limit test results of the residues in the real samples determined using TLC-SERS were verified via UPLC–MS/MS; A, B, C, D, E, F, G, and H were separated via gradient elution using a Kromasil C_18_ column (100 × 2.1 mm × 1.8 μm) with a mobile phase at a flow rate of 0.2 mL/min. The elution procedure was as follows: 0~3 min, 78% A, 20% B, 2% C; 3~6 min, 75% A, 20% B, 5% C; 6~8 min, 70% A, 20% B, 10% C; 8~13 min, 40% A, 20% B, 40% C; 13~13.1 min, 40% A, 20% B, 40% C; 13.1~16 min, 78% A, 20% B, 2% C; 16 min, 78% A, 20% B, 2% C (A: 0.1% formic acid solution containing 5.0 mmol/L ammonium acetate; B: methanol; C: acetonitrile). The column temperature was the same as the room temperature. Positive-ion ESI in the MRM mode was used to monitor the precursor ion→product ion transitions of m/z 360→316 (A), 332→288 (B), 362→318 (C), 370→326 (D), 393→349 (E), 321→303 (F), 376.2→332.2 (G), and 361→343.2 (H). The CE values of the eight quinolones were 20 V (A), 9 V (B), 19 V (C), 19 V (D), 18 V (E), 20 V (F), 21 V (G), and 17 V (H); and the DP value of these quinolones was 80 V.

### 3.3. Preparation of Silver Nanoparticles

To obtain the SERS values of the eight quinolones, we prepared silver nanoparticles, which we also called nanometer silver solution or active SERS substrates in this paper. We dissolved 56 mg of silver nitrate in 150 mL of water to obtain a mixed solution by evenly adding 4 mL of 1% sodium citrate solution to the silver nitrate solution. The nanometer silver solution was prepared by heating the mixed solution in a microwave oven until it boiled for 5 min and then cooling it to room temperature. This solution was stored at 4 °C to ensure its stability.

### 3.4. Preparation of the Reference Solutions and Their Mixtures

For the reference solutions, for example, to prepare the quinolone A solution, we dissolved an appropriate amount of the reference substance (A) in anhydrous ethanol to a concentration of 1.00 mg/mL. Following the same method, the reference solutions of the other quinolones (B, C, D, E, F, G, and H) were prepared at a concentration of 1.00 mg/mL.

For the mixed reference solution 1, to detect the degree of separation of the eight quinolones with TLC, a mixed solution was prepared by placing 1.00 mL of each of the above reference solutions (A, B, C, D, E, F, G, or H) in the same container. After the mixture of solutions was dried in a water bath (85 °C), it was redissolved in 1.00 mL of anhydrous ethanol to obtain the mixed reference substance solution 1.

To prepare the mixed reference substance solution 2, according to the MRL of quinolones in food, we dissolved an appropriate amount of the eight quinolones in the same portion of anhydrous ethanol. In the solution, the concentrations of quinolones A, B, C, D, E, F, G, and H were 400.0, 400.0, 8.0, 20.0, 20.0, 20.0, 20.0, and 20.0 ng/mL, respectively.

### 3.5. Preparation of the Sample Solutions

To accurately obtain the SERS of quinolones in food, 2.00 g of the food sample, 10 g of Na_2_SO_4_ powder, and 10 mL of anhydrous ethanol were placed in the same centrifuge tube. The residues were extracted from the food substrate via sonication for 15 min. After centrifugation at 4000 rpm for 5 min, the supernatant, which may contain the quinolones was passed through a filter membrane (0.22 μm) to obtain a filtrate. The filtrate was concentrated in a water bath (85 °C) to approximately 1 mL and then transferred to a chromatographic vial. All the solvent in the vial was evaporated in a water bath; the residues of the food were redissolved in 500.0 μL of anhydrous ethanol to obtain the sample solution.

### 3.6. TLC

TLC is a simple and fast separation technique. The eight quinolones were preliminarily separated as follows: We separately spotted 10.0 µL of the reference substance solutions and mixture reference substance solution 1 onto a GF_254_ thin-layer plate (10 cm × 10 cm) at a distance of 1 cm from the bottom. When the plate was eluted to a distance of 8 cm via dichloromethane:methanol (5:1) in a glass container, it was removed from the container, and the agent R_f_ on the plate was naturally evaporated. Irradiating at 254 nm, the main spots on the thin-layer chromatogram were observed. Using fleroxacin (D) as the reference, the relative R_f_ values of A, B, C, D, E, F, G, and H were separately measured using these spots.

### 3.7. TLC-SERS

In this study, we focused on developing a rapid and specific limit test method called TLC-SERS to identify the levels of eight quinolone residues in food. Using TLC, 10.0 µL each of the fleroxacin (D) reference solution and the mixed reference solution 2 were separately dropped onto the same thin-layer plate. Under ultraviolet-ray irradiation at 254 nm, we clearly observed a spot on the thin-layer chromatograph of fleroxacin. Because the concentrations of the eight quinolone components in the mixed reference solution 2 were lower than their LOD on the TLC; so, no spot was observed in the chromatogram of the mixed solution. Therefore, the eight components’ positions were separately marked using their R_f_ values. These R_f_ values were calculated using their relative R_f_ and the R_f_ of fleroxacin. Next, silver nanoparticles (the active SERS substrate) were gently added to each of these marked positions using a microinjector. Only 1 µL of silver nanoparticles was added at each position each time. After allowing the nanoparticles to dry naturally, this operation was repeated 6 times to obtain a uniformly distributed circle with a diameter of approximately 5 mm on the TLC plate. When the center of the circle was irradiated using a laser at 532 nm from the Raman spectrometer, the SERS of the quinolones was directly obtained. In addition, to identify the Raman signal of the active SERS substrate, the blank position with an R_f_ value that was the same as that of the eight quinolones was marked and detected following the same method. Under normal circumstances, we obtained the SERS of the eight quinolones in the mixed reference solution 2 using TLC-SERS; no obvious Raman signal was observed for the blank position.

Following the above method, when the sample solution was dropped onto the thin-layer plate and the residues were separated via TLC, the SERS values of the residual quinolones in food were also obtained. When the SERS of the components in the food was consistent with the that of the corresponding reference substance in the mixed reference solution 2, we identified the corresponding quinolone residues in the foods. Based on this, if all characteristic peaks of the component were stronger than those of the corresponding reference, we identified the levels of the residual quinolones in the food as exceeding the MRL, indicating that the quality of the food was unacceptable. Otherwise, it indicated that the levels of the residual quinolones in the food did not exceed their MRLs.

## 4. Conclusions

In this study, a method (TLC-SERS) for detecting the levels of quinolone residues in aquatic products and other animal foods was established. TLC-SERS features high sensitivity, strong specificity and selectivity, reliable accuracy, and good stability. In addition, compared with the existing methods, the method is simpler and faster.

The results of this experiment show that the SERS values of the quinolones acquired with TLC-SERS strongly correlated with the Raman spectra of the corresponding reference substance. By comparing the relative intensities and the Raman shifts of their characteristic peaks, the SERS values of the different quinolones were substantially different. By comparing the results of determining the compounds in three solutions (reference substance solution, simulated positive sample solution, and negative sample solution), we found that the components in the food matrices did not interfere with the test results of the residues. By measuring the relative changes in the heights of the four characteristic peaks (*γ*_C=C_, *β*_CH2_, *β*_CH3_, and γ_C-N_) of the same sample at different times, a RSD of ≤4.9% was obtained for each quinolone compound. By measuring the different concentrations of the quinolone reference substance solutions, we found that the LOD was lower than or equal to its MRL for each quinolone compound. Through a comparison of the Raman spectral characteristic peak intensity per unit mass with that of the SERS, we found that the EF ≥ 1.1 × 10^4^ for each quinolone compound. In addition, the limit test results using twenty real samples proved that, except for enrofloxacin (A) and ciprofloxacin (B) found in two batches of samples, no quinolone residues were detected in the other samples. Only in one batch of samples did the content of enrofloxacin (A) exceed its MRL, which was consistent with the results produced using the authoritative analysis method (UPLC–MS/MS).

In conclusion, the proposed method provides a new way to rapidly test the levels of harmful residues in foods.

## Figures and Tables

**Figure 1 molecules-28-06473-f001:**
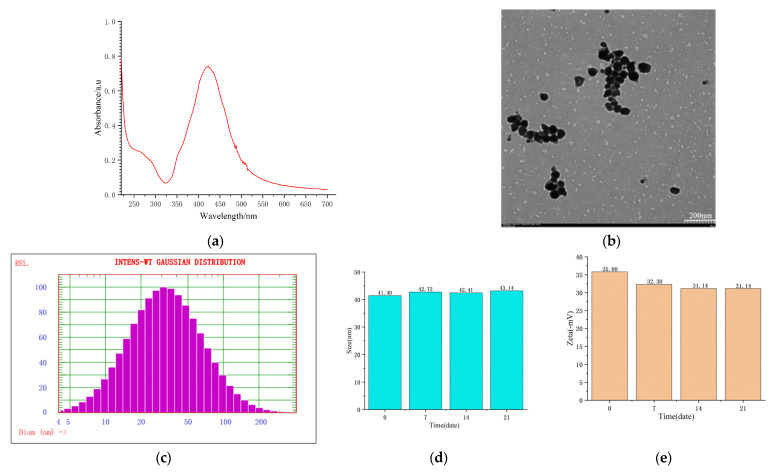
Characterization and stability of the SERS of the active substrates. (**a**) Ultraviolet absorption spectrum; (**b**) appearance; (**c**,**d**) particle size; and (**e**) zeta potential.

**Figure 2 molecules-28-06473-f002:**
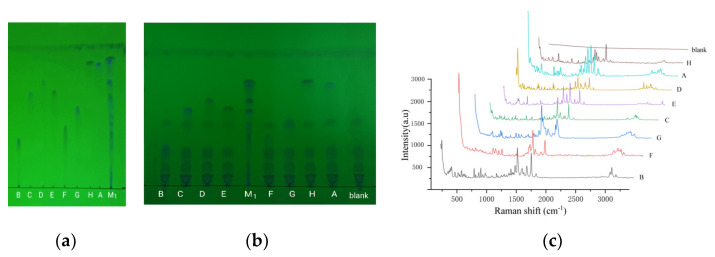
TLC and Raman spectra diagram of eight quinolones. (**a**–**c**) TLC diagram of 8 reference solutions, TLC diagram of mixture of the reference and food, and Raman spectra diagram of 8 quinolones, respectively. A, B, C, D, E, F, G, H, and M_1_ represent the eight quinolone reference solutions and the mixed reference solution 1, respectively; “blank” represents the liquid product solution without any quinolones.

**Figure 3 molecules-28-06473-f003:**
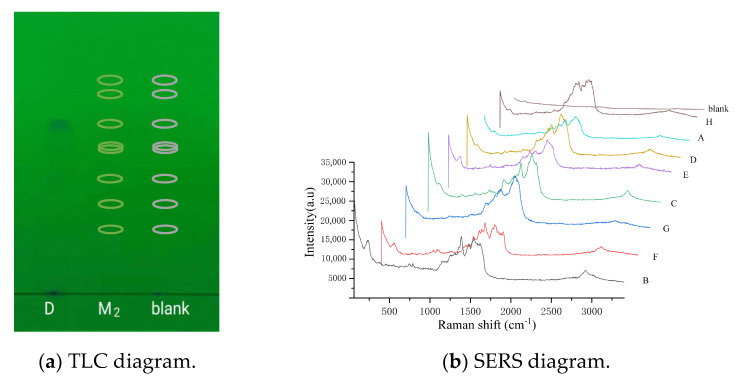
TLC and SERS diagrams of the eight quinolones in the mixed reference substance solution 2. According to the TLC diagram, D represents the fleroxacin solution, and the spot was from fleroxacin; M_2_ represents the mixed reference solution 2, with eight quinolones (H, A, D, E, C, F, G, F, and B) in the marked positions in the chromatogram; “blank” represents anhydrous ethanol, with no quinolones in the marked positions in the chromatogram. In the SERS diagram, A, B, C, D, E, F, G, and H represent the different quinolones in mixed solution 2; “blank” represents anhydrous ethanol.

**Figure 4 molecules-28-06473-f004:**
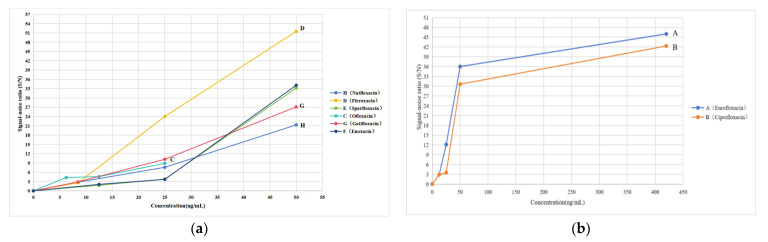
LODs of eight quinolones obtained via TLC-SERS. H, D, E, C, G, and F in the (**a**) represent nadifloxacin, fleroxacin, sparfloxacin, ofloxacin, gatifloxacin, and enoxacin, respectively. A and B in the (**b**) represent enrofloxacin and ciprofloxacin, respectively.

**Figure 5 molecules-28-06473-f005:**
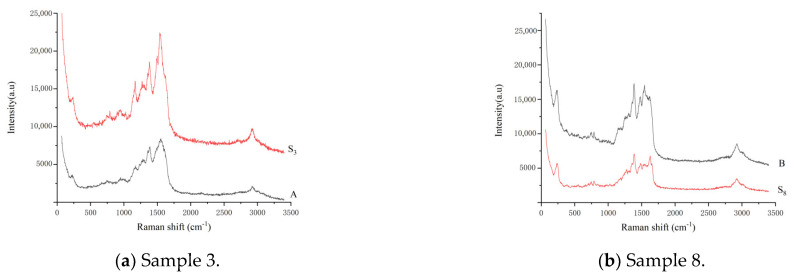
SERS of samples determined with TLC−SERS. (**a**) S_3_, sample 3, liquid product; A: enrofloxacin. (**b**) S_8_, sample 8, other animal foods; B: ciprofloxacin.

**Table 1 molecules-28-06473-t001:** Raman shift and EFs of the characteristic peaks of the eight quinolones.

Structure/ Relative R_f_	Raman Shift (cm^−1^) of Blank Substrate/ Relative Peak Intensity	Raman Shift (cm^−1^) of SERS of Active Substrate/Relative Peak Intensity	Functional Group	EF
Nadifloxacin (H) 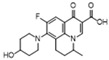 R_f_ = 1.22	2964~2861 (2 peaks)	2991~2846 (1 peak)	Common peak: _ν=CH,ν-CH2,ν-CH3_	
	1721~1147 (9 peaks)	1610~1157 (8 peaks)	Characteristic peak:	
	1721/0.34		_νC=O_	
	1621/1.65	1610/1.01	_νC=C_ from phenyl rings	1.1 × 10^6^
	1403/1.00	1398/1.00	_νC=C_ from phenyl rings	8.8 × 10^5^
	1363/1.24	1358/0.96	β_CH2_, β_CH3_	2.6 × 10^6^
	1147/0.20	1157/0.32	_νC-N_	1.4 × 10^6^
Enrofloxacin (A) 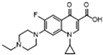 R_f_ = 1.16	3094~2833 (5 peaks)	2995~2823 (1 peak)	Common peak: _ν=CH,ν-CH2,ν-CH3_	
	1743~1129 (9 peaks)	1600~1173 (4 peaks)	Characteristic peak:	
	1743/0.08		_νC=O_	
	1629/0.29	1600/0.94	_νC=C_ from phenyl rings	3.1 × 10^4^
	1400/1.00	1389/1.00	_νC=C_ from phenyl rings	1.1 × 10^4^
	1350/0.71	1281/0.77	β_CH2_, β_CH3_	1.3 × 10^5^
	1129/0.12	1173/0.65	_νC-N_	2.0 × 10^5^
Fleroxacin (D) 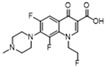 R_f_ = 1.00	3057~2800 (6 peaks)	3037~2806 (1 peak)	Common peak: _ν=CH,ν-CH2,ν-CH3_	
	1791~1147 (7 peaks)	1595~1255 (5 peaks)	Characteristic peak:	
	1791/0.21		_νC=O_	
	1634/0.94	1595/1.12	_νC=C_ from phenyl rings	1.2 × 10^6^
	1387/1.00	1379/1.00	_νC=C_ from phenyl rings	2.7 × 10^5^
	1332/0.65	1298/0.77	β_CH2_, β_CH3_	1.2 × 10^6^
	1147/0.21	1255/0.64	_νC-N_	6.3 × 10^5^
Sparfloxacin (E) 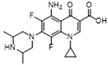 R_f_ = 0.90	3095~2835 (6 peaks)	2997~2823 (1 peak)	Common peak: _ν=CH,ν-CH2,ν-CH3_	
	1721~1179 (9 peaks)	1621~1176 (5 peaks)	Characteristic peak:	
	1721/0.32		_νC=O_	
	1632/1.38	1621/1.21	_νC=C_ from phenyl rings	6.9 × 10^5^
	1367/1.00	1373/1.00	_νC=C_ from phenyl rings	1.8 × 10^5^
	1296/1.69	1279/1.03	β_CH2_, β_CH3_	1.1 × 10^5^
	1179/0.70	1176/0.71	_νC-N_	4.8 × 10^5^
Ofloxacin (C) 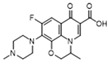 R_f_ = 0.89	3002~2763 (6 peaks)	3003~2810 (1 peak)	Common peak: _ν=CH,ν-CH2,ν-CH3_	
	1720~1147 (9 peaks)	1612~1151 (5 peaks)	Characteristic peak:	
	1720/0.16		_νC=O_	
	1632/0.85	1612/0.98	_νC=C_ from phenyl rings	4.0 × 10^6^
	1401/1.00	1394/1.00	_νC=C_ from phenyl rings	1.0 × 10^6^
	1329/0.29	1300/0.68	β_CH2_, β_CH3_	5.9 × 10^6^
	1147/0.23	1151/0.57	_νC-N_	3.0 × 10^6^
Gatifloxacin (G) 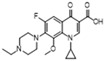 R_f_ = 0.74	3081~2846 (6 peaks)	3026~2806 (1 peak)	Common peak: _ν=CH,ν-CH2,ν-CH3_	
	1620~1185 (9 peaks)	1562~1151 (6 peaks)	Characteristic peak:	
	1620/0.93	1562/1.33	_νC=C_ from phenyl rings	2.7 × 10^6^
	1350/1.00	1357/1.00	_νC=C_ from phenyl rings	2.4 × 10^6^
	1328/1.57	1286/0.85	β_CH2_, β_CH3_	3.1 × 10^6^
	1185/0.24	1151/0.65	_νC-N_	2.6 × 10^6^
Enoxacin (F) 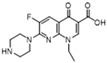 R_f_ = 0.56	3055~2878 (4 peaks)	3032~2823 (1 peak)	Common peak: _ν=CH,ν-CH2,ν-CH3_	
	1674~1239 (7 peaks)	1649~1161 (7 peaks)	Characteristic peak:	
	1627/0.63	1649/0.74	_νC=C_ from phenyl rings	1.1 × 10^6^
	1409/1.00	1414/1.00	_νC=C_ from phenyl rings	1.0 × 10^5^
	1355/0.48	1340/0.78	β_CH2_, β_CH3_	4.0 × 10^5^
	1239/0.10	1265/0.60	_νC-N_	1.0 × 10^6^
Ciprofloxacin (B) 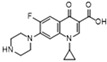 R_f_ = 0.44	3089~2990 (3 peaks)	2995~2823 (1 peak)	Common peak: _ν=CH,ν-CH2,ν-CH3_	
	1711~1162 (8 peaks)	1621~1151 (8 peaks)	Characteristic peak:	
	1711/0.16		_νC=O_	
	1628/0.82	1621/0.85	_νC=C_ from phenyl rings	7.0 × 10^4^
	1389/1.00	1386/1.00	_νC=C_ from phenyl rings	9.9 × 10^3^
	1277/0.31	1307/0.64	β_CH2_, β_CH3_	5.7 × 10^4^
	1162/0.14	1151/0.47	_νC-N_	9.6 × 10^4^

ν represents stretching vibration; β represents in-plane bending vibration. The relative intensity of the peaks was obtained using the ratio of the absolute intensity to that of the reference peak, and the relative intensity of the reference peak was equal to 1.

**Table 2 molecules-28-06473-t002:** Comparison of LODs and MRLs of the eight quinolone residues in food.

Quinolone	MRL (μg/kg)	LOD (μg/kg)	MRL (ng/mL)	LOD (ng/mL)
Nadifloxacin (H)	5.0	2.2	20.0	9.0
Enrofloxacin (A)	100.0	3.2	400.0	12.6
Fleroxacin (D)	5.0	2.2	20.0	8.9
Sparfloxacin (E)	5.0	4.8	20.0	19.0
Ofloxacin (C)	2.0	2.0	8.0	8.0
Gatifloxacin (G)	5.0	2.2	20.0	8.7
Enoxacin (F)	5.0	4.8	20.0	19.0
Ciprofloxacin (B)	100.0	3.2	400.0	12.6

**Table 3 molecules-28-06473-t003:** Information on 20 different food samples.

Liquid Samples		Animal Food	
Sample 1	Grass carp	Sample 11	Chicken
Sample 2	Perch	Sample 12	Pork
Sample 3	Shrimp	Sample 13	Beef
Sample 4	Treasure fish	Sample 14	Mutton
Sample 5	Salmon	Sample 15	Chicken liver
Sample 6	Carp	Sample 16	Pork liver
Sample 7	Saury	Sample 17	Beef liver
Sample 8	Silver carp	Sample 18	Mutton liver
Sample 9	Crucian carp	Sample 19	Eggs
Sample 10	Squid	Sample 20	Milk

## Data Availability

Not applicable.

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
