# Peer review of "Rapid Limit Test of Eight Quinolone Residues in Food Based on TLC-SERS, a New Limit Test Method"

_molecules, 2023, doi:10.3390/molecules28186473_

Round 1
Reviewer 1 Report
1. In the manuscript, the authors have been established the TLS-SERS method for rapid limit test of the eight quinolones residue in the food. The overall content of the manuscript is full, and the expression is smooth, which has research value and reference significance. At the part of ‘Apparatus and Conditions’, the study used the UPLC-MS/MS method to do some determination. What are the CE values and DP values?
2.Please check if the functional group attribution in Table 1 corresponds one by one.
3.Double check the typos in manuscript, Line 64, sepseparated
4. Please check the Raman shift values of some functional groups in Table 1.
5. Please provide ion flow chromatograms of relevant compounds in the auxiliary materials.
1. In the manuscript, the authors have been established the TLS-SERS method for rapid limit test of the eight quinolones residue in the food. The overall content of the manuscript is full, and the expression is smooth, which has research value and reference significance. At the part of ‘Apparatus and Conditions’, the study used the UPLC-MS/MS method to do some determination. What are the CE values and DP values?
2.Please check if the functional group attribution in Table 1 corresponds one by one.
3.Double check the typos in manuscript, Line 64, sepseparated
4. Please check the Raman shift values of some functional groups in Table 1.
5. Please provide ion flow chromatograms of relevant compounds in the auxiliary materials.
Reviewer 2 Report
In this paper, the authors tried to develop a thin-layer chromatography
(TLC)-surface-enhanced Raman spectroscopy (SERS) method to rapidly determine multiple residues including enrofloxacin, ciprofloxacin, ofloxacin, fleroxacin, sparfloxacin, enoxacin, gatifloxacin and nadifloxacin in food. The combination of TLC endowed the SERS analysis method with the separation function, so the detection of multiple residues in complex food could be potentially achieved. The idea is feasible. However, there are several concerns needing address, and some crucial evidences or data regarding with the analytical methodology are missing. In current status, it’s afraid that the paper is not suitable for formal publication. Here are some suggestion for the paper.
1. The ultimate purpose of the established method is for qualification or quantification. Please clarify the point clearly in the paper. SERS has been used as a powerful analytical technology for many years. In recent decay, the rapid quantification by SERS has been the hot spot in this field. If the work only focused on the qualification analysis, the novelty is limited.
2. In the introduction section, please clearly state the significance or advantage of the proposed SERS method for these eight residues in comparison with other conventional methods. Chromatography methods, especially chromatography hyphenated with mass spectrometry, can provide powerful separation and analysis results for hundreds of residues in one run with good accuracy and sensitivity. Thus, the priority of the proposed method should be systematically evaluated and compared, and it’s suggested to summarize the comparison information in a table, demonstrating the LOD, precision, time-consumption, etc.
3. The characteristic SERS peaks in all SERS spectra should be labeled. And the peak for quantification should be marked clearly.
4. The enhancement substrate for this work was Ag nanoparticles. Why not Au nanoparticles? Au nanoparticles generally demonstrate better stability compared with Ag nanoparticles. Please compare the SERS performance of these two substrates. Moreover, the time range for stability test in section 3.7 is within 8 hours which is not satisfied with real sample analysis. It’d better to test the stability over one week at least, because the enhancement substrates after preparation will be usually stored at fridge for consequent usage during the following days.
5. Enhancement factor (EF) is a crucial parameter for SERS analysis, but it is not provided in the work. Please add EFs for each target residue in the paper.
6. In Fig. 5, the calibration curves were provided, which suggested the quantification range of the proposed method. However, the calibration model could not meet the requirement of quantification in fact. There was no linearity based on the calibration model. Most of calibration curves were based on 3 or 4 data points, but the calibration range should be based on 5 points at least. And there were no any error bars in the data points. Did the replicated measurements for each point conducted? The precision of the method should be evaluated based on the replicated measurements and corresponding relative standard deviations.
7. In table 2, several LOD values were higher than the low points of concentration range, which was unreasonable. Please doubly check the data.
8. The selectivity and anti-stability is also crucial evidence for the feasibility of the proposed method. Please provide corresponding data.
9. For real sample analysis, there was not clear and detailed information regarding the food sample and sample preparation procedure. What kind of food sample has been analyzed by the proposed method? How to obtain the testing solution for TLC-SERS analysis? Please amend the corresponding information.
10. Ag nanoparticles were added in the specific spots on the TLC plate before or after TLC process of sample solution? How to guarantee the homogeneous distribution of the nanoparticles on the TLC plate and interaction with the targets, considering the coffee-ring effect?
11. The paper organization should be further improved to condense the crucial information but delete the redundant content. Some expressions should be reconsidered to be more formal.
12. There are typo and grammatical errors in the paper. Please polish the language before resubmission.
The paper organization should be further improved to condense the crucial information but delete the redundant content. Some expressions should be reconsidered to be more formal. There are typo and grammatical errors in the paper. Please polish the language before resubmission.
Round 2
Reviewer 2 Report
The paper has been revised according to the reviewing comments. It's suitable for acceptance in this journal currently.